# Red-GAN: Attacking class imbalance via conditioned generation. Yet another medical imaging perspective.

**Ahmad B. Qasim**[*]                                           AHMAD.QASIM@TUM.DE
**Ivan Ezhov**[*]                                                IVAN.EZHOV@TUM.DE
**Suprosanna Shit**                                             SUPROSANNA.SHIT@TUM.DE
**Oliver Schoppe**                                              OLIVER.SCHOPPE@TUM.DE
**Johannes C. Paetzold**                                        JOHANNES.PAETZOLD@TUM.DE
**Anjany Sekuboyina**                                           ANJANY.SEKUBOYINA@TUM.DE
**Florian Kofler**                                              FLORIAN.KOFLER@TUM.DE
**Jana Lipkova**                                                JANA.LIPKOVA@TUM.DE
**Hongwei Li**                                                  HONGWEI.LI@TUM.DE
**Bjoern Menze**                                                BJOERN.MENZE@TUM.DE
*TranslaTUM and Department of Informatics, Technical University of Munich, Germany*

## Abstract

Exploiting learning algorithms under scarce data regimes is a limitation and a reality of the medical imaging field. In an attempt to mitigate the problem, we propose a data augmentation protocol based on generative adversarial networks. We condition the networks at a pixel-level (segmentation mask) and at a global-level information (acquisition environment or lesion type). Such conditioning provides immediate access to the image-label pairs while controlling global class specific appearance of the synthesized images. To stimulate synthesis of the features relevant for the segmentation task, an additional passive player in a form of segmentor is introduced into the adversarial game. We validate the approach on two medical datasets: BraTS, ISIC. By controlling the class distribution through injection of synthetic images into the training set we achieve control over the accuracy levels of the datasets' classes[1].

## 1. Introduction

The recent rise in development of neural networks is a consequence of multiple factors, such as adaptation of graphics processing units to general-purpose computing and algorithmic maturity. Concurrently, access to abundant annotated data has been imperative for parametric optimization of networks. Training under a scarce data regime leads to sub-optimal network parameters and affects generalization ability. The first strategy to remedy situations when the availability of annotated data is limited is to employ data augmentations.

Efficient data augmentation should result in increased variability of the training set. In case the data possesses invariance to a certain transformation the dataset can be enriched with correspondingly transformed original data. Various transformations, e.g. scal-

---

[*] Contributed equally

1. The code is available at https://github.com/IvanEz/Red-GAN

ing, rotation, translation, etc., are typically used for augmenting imaging data (Shorten and Khoshgoftaar, 2019). However, for multi-class images the intricate geometrical interrelation between constituting parts can constrain the application of the simplistic image manipulations. Several previous works demonstrated the possibility to exploit generative adversarial networks, GANs (Goodfellow et al., 2014; Schmidhuber, 2020), for the augmentation task (Antoniou et al., 2017; Pal and Balasubramanian, 2018; Zhang et al., 2018a).

As shown in (Antoniou et al., 2017; Mariani et al., 2018; Salimans et al., 2016; Odena et al., 2016; Zhang et al., 2018a; Li et al., 2017; Vandenhende et al., 2019), augmentation of a training set with samples generated by label-conditional GANs can result in increased classification performance. The authors of (Salimans et al., 2016; Odena et al., 2016; Zhang et al., 2018a) construct an objective function for the discriminator, allowing it to play a role of a classifier for semantic classes in a semi- (Salimans et al., 2016; Odena et al., 2016) and fully-supervised (Zhang et al., 2018a) fashion, in addition to the fake versus real discrimination role. Such objective enforces generation of realistic as well as class specific samples. In (Li et al., 2017; Vandenhende et al., 2019), a third player is introduced into the two-player game to control the semantics of generated samples independently from the discriminator.

In the medical imaging context, the GAN-based augmentation has been applied to classification (Frid-Adar et al., 2018; Gupta et al., 2019) and segmentation problems (Mok and Chung, 2018; Thomas et al., 2017; Zhang et al., 2018b; Bowles et al., 2018; Rezaei et al., 2017; Abhishek and Hamarneh, 2019). A wider review can be found in (Kazeminia et al., 2018; Yi et al., 2019). In (Thomas et al., 2017; Bowles et al., 2018) a generator model is learned to map a random vector to real distribution, which is a joint distribution (i.e. concatenation) of an image and corresponding segmentation. While in (Bowles et al., 2018) the augmentation led to an improvement in the segmentation of brain lesion and anatomy, the authors of (Thomas et al., 2017) observed symptoms of mode collapse (Arora and Zhang, 2017), manifested in similar appearance of the generated X-ray images of thorax conditioned on different random vectors. This leads to low variability of the augmented images set. In (Mok and Chung, 2018), for the synthesis of brain tumor images, the network design is adopted from Pix2PixHD (Wang et al., 2017), which is composed of a generator conditioned on a segmentation mask. A network equipped with such conditioning should be less prone to the mode collapse, given that spatial diversity of the images is controlled by the mask. However, all these studies do not differentiate between the varying global property inherent to the data. For example, in cases when the data are obtained from different centers, the appearance of medical scans varies between the centers (e.g. due to varying scanner type, acquisition protocol, etc.). Neglecting this can affect the networks' performance, since the segmentation mask-image pairs are far from unique.

In this work, we propose a way to address the problem by conditioning the generation on the *global* information – a *global class* such as acquisition set-up or lesion type – in addition to the *local* one (segmentation masks). Such conditioning allows to mitigate the "synthesis dilemma" for the task of data augmentation: to train a generative network that produces high quality images one needs abundant real data, but if one has abundant data, then this amount is already enough for a segmentation network to learn a "generalizable" input-output mapping and an extra addition of synthetic images does not improve notably the performance. In our case, the generative network conditioned on the local-global infor-

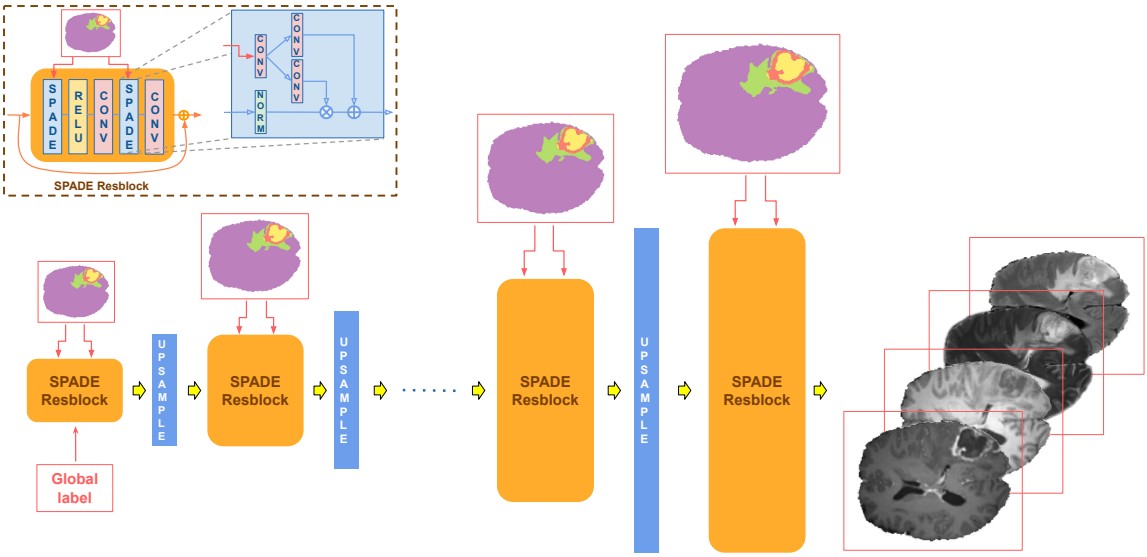

Figure 1: Generator design.

mation makes use of the representation learned on the whole dataset for the synthesis of a particular *global class*. Thus, for datasets with *global* class imbalance we can reach high quality of the synthesis even for sparse classes.

The contribution of the paper is as follows:

(a) We propose an adversarial framework conditioned on both the local and global information. This provides access to the mask-image pairs while controlling class specific appearance of the generated images. We incorporate a third player in the adversarial game to play in favour of the downstream segmentation task. We coin the method as Red-GAN. (b) We validate the framework on two different medical datasets (BraTS (Menze et al., 2015) and ISIC (Codella et al., 2019)) by performing a series of tests wherein class specific synthetic samples are injected into the original set. By controlling the class distribution we achieve a control over the segmentation performance for the cohort of the datasets' classes.

## 2. Method

Our goal of the image generation is augmentation of the original training set for the downstream segmentation task. For this, we make use of the existing masks to synthesize new images. We base our method on the state-of-the-art conditional generative network, SPADE (Park et al., 2019), that allows to generate synthetic images directly from label masks. The core of the SPADE, is the generator network composed of residual blocks with up-sampling, Fig. 1. In contrast to Pix2PixHD (Wang et al., 2017) the segmentation mask is spatially adapted to be fed into each block in order to more efficiently preserve semantic information through the whole depth of the generator.

Conventional GAN design implies two players, namely the generator and discriminator, competing between each other. In our work we introduce a third passive player - a seg-

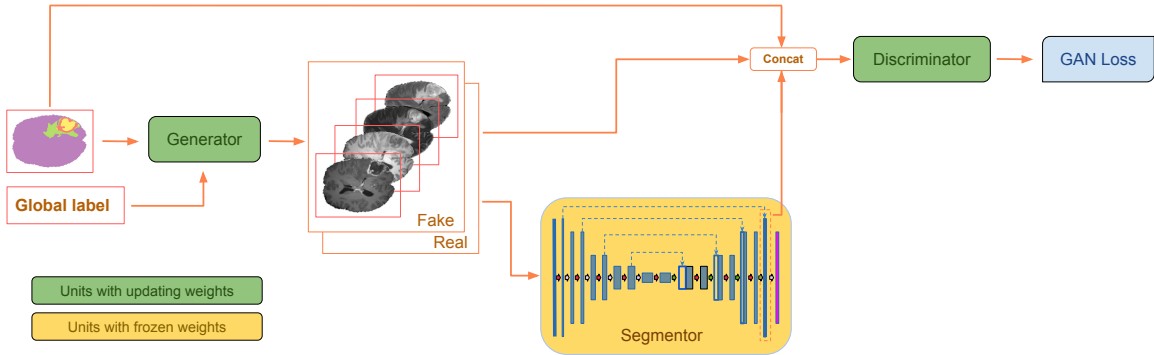

Figure 2: Red-GAN.

mentor, which is trained on the same dataset before the adversarial game begins and is frozen during the game. Both the synthetic images (stacked slices from all modalities in the case of BraTS) produced by the generator and the real images are separately passed through the segmentor. As input, the discriminator receives the mask, the image slices (real or synthesised), and the feature representations obtained by passing these image slices through the U-Net. Refer to Fig. 2 for an illustration. The intuition behind inclusion of the third player is as follows. If we train a generator from scratch, then it learns a general representation of images. Since we want to use the synthetic images for segmentation task, we want to ensure the images lie within close proximity to the real images in the latent representation, based on which the segmentor makes its decision.

### 2.0.1. IMPLEMENTATION.

The generator is comprised of 7 SPADE resnet blocks coupled together. A SPADE resnet block, is a combination of the basic operations, Fig. 1. Except for the first and the last block, an upsampling layer is added after each block, hence 5 upsampling layers in total are added to the generator. At first, the input segmentation masks are downsampled to 8×8 from 256×256, then passed through the generator network. After iterative upsampling by doubling image size, the output of the generator is 256×256. For the BraTS dataset we modified the generator part to output 2D images from all available imaging modalities (instead of the original three channel output for color images). The class conditioning is implemented analogous to (Denton et al., 2015) wherein we used an embedding layer followed by a fully connected layer reshaping the embedding to the 1024x8x8 size. The latter is then concatenated with the mask inside the first SPADE block. The concatenation occurs at the place where the mask is transformed by the block functional units to the same dimension size 1024×8×8.

The discriminator is based on the PatchGAN architecture (Phillip et al., 2017; Li et al., 2019). The discriminator has a multiscale architecture consisting of two sub-discriminators. Each sub-discriminator has 3 convolutional layers stacked together. After a forward pass through each sub-discriminator, an average pooling layer is added to downsample the resultant tensor. Spectral normalization is applied to all convolutional layers in the generator

and the discriminator. We used Hinge loss and feature matching loss analogous to the SPADE design. The learning rate of 1e-4 is used for the generator and 4e-4 is used for the discriminator. The Adam (Kingma and Ba, 2014) optimizer is used with $\beta_1 = 0$ and $\beta_2 = 0.9$. The Red-GAN is trained for 80 epochs.

As a segmentor, we used a U-Net architecture (et al., 2015) and a combination of Jaccard and Cross-Entropy as a loss (Yakubovskiy). For the U-Net encoder, we use the Resnet-34 architecture (He et al., 2015). The decoder consists of 5 decoder blocks and a final convolutional layer. Each decoder block is a convolutional layer with ReLU activation. The network is trained from scratch, with no pre-trained weights, using Adam optimizer. The same architectures were used inside the GAN and for the segmentation task. The segmentor is trained for 100 epochs. We did not apply any other augmentations because our aim was to test the viability of our contributions, i.e. the segmentor and global conditioning, on the downstream segmentation task without introducing other factors which can affect the segmentation results.

## 3. Results

### 3.0.1. Data.

We validate the proposed framework on two widely studied medical datasets:

- Multimodal Brain Tumor Image Segmentation (BraTS) (Menze et al., 2015)

- International Skin Imaging Collaboration: Melanoma Project (ISIC) (Codella et al., 2019)

The BraTS2019 dataset consists of MRI scans of glioma patients with manual segmentation by an expert board. The total set comprises 335 MRI images, among which 259 are high-grade glioma patients and 76 are low-grade glioma (we sliced 16.5K 2D images out of the 3D volumes for our task). For each patient, scans from 4 imaging modalities are available: T2-weighted, T2 Fluid Attenuated Inversion Recovery (FLAIR), T1, and T1-weighted. The images annotations are the GD-enhancing tumor, the peritumoral edema, the necrotic and non-enhancing tumor core. The dataset was obtained from multiple medical centers and we observe notable difference in the appearance of the scans across the centers. Thus, we use the the center ID available from the naming of the data (e.g. CBICA, TMC, etc.) as the class conditioning.

The ISIC2018 dataset contains over 13000 dermoscopic images of skin lesions. We used its subset of $\sim$2600 that was provided for the ISIC segmentation challenge. The challenge data are annotated by clinical experts to outline the lesion area and also includes metadata as a type of the lesion. For the class conditioning, we used the skin lesion type, e.g. Melanoma, Seborrheic Keratosis, Nevus.

As depicted in Fig.3 (upper row), both datasets exhibit class imbalance, which we aim to eliminate by augmenting the datasets with class-specific synthetic samples.

### 3.0.2. Experiments.

First, we quantitatively show an effect of the third-player inclusion by training a 2D U-Net only on synthetic images produced by the original and proposed GANs while testing on

| | SPADE-GAN | SPADE-GAN with the third-player |
|---|---|---|
| BraTS | 0.6479 (0.0127) | 0.6598 (0.0152) |
| ISIC | 0.5936 (0.0283) | 0.6169 (0.0361) |

Table 1: Dice scores for segmentation using synthetic images produced by the SPADE-GAN and using the SPADE equipped with the third-player (no class conditioning is used). For the GAN training we used 90% percent of the total amount of 2D slices/images. Testing is performed on 10% of real data from original dataset. For the synthesis we used masks from the training set. The results for the mean are obtained via 3 cross-fold validation.

real data from the original dataset, Tab. 1. With this we aim to probe the quality of the synthetic set. We observe an improvement of the accuracy (multi-class DICE score) for the proposed architecture with feature level discrimination compared to the original SPADE. For the BraTS dataset, the p-value is equal to 0.033, and for ISIC - 0.028 using the paired Wilcoxon signed rank test. The performance gain towards SPADE-GAN in Tab. 1 is only due to the introduction of the third player into the adversarial game. All the training and architectural details were kept the same and the global class conditioning is not yet used.

Next, we employ the proposed Red-GAN with both the third-player and class conditioning. We perform two series of experiments in which we train a 2D U-Net on original dataset augmented with synthetic samples according to the following strategies:

I. **Single class augmentation.** We inject synthetic images that are generated for a particular class from all masks in the training set except the ones belonging to the class.
II. **Balanced augmentation.** We separately synthesize images that are generated according to (I) for all classes and inject them in the dataset.

The first strategy biases the class distribution with respect to the synthesised class making its number of samples equal to the size of the whole original dataset. This strategy should allow us to probe how specific the synthetic images to the class on which they were conditioned. The second makes the distribution balanced by increasing the size of each class to the original dataset size. In Fig. 3, we compare both of them with a baseline that is a U-Net trained without the GAN-based augmentation (depicted in the grey dots). We observe that by using the strategy (I), a strong accuracy increase compared to the baseline is achieved for the injected class (5% for the BraTS "TCIA05" class and 4% for the ISIC "Keratosis" class). For most of the other classes there is a smaller increase or even decrease of the Dice score. This suggests that the generated images posses the desired property of being specific to the conditioned class. Plots for other single class injection experiments as well as examples of the synthesized images are provided in the appendix.

As depicted in the bottom row, by using the balanced augmentation (strategy II), we can achieve increase of the DICE score for most of the sparse classes (up to 5% for BraTS and up to 2% percent for ISIC). We note that difference in visual appearance between various skin lesions is clearly greater compared to the difference between the MRI images of

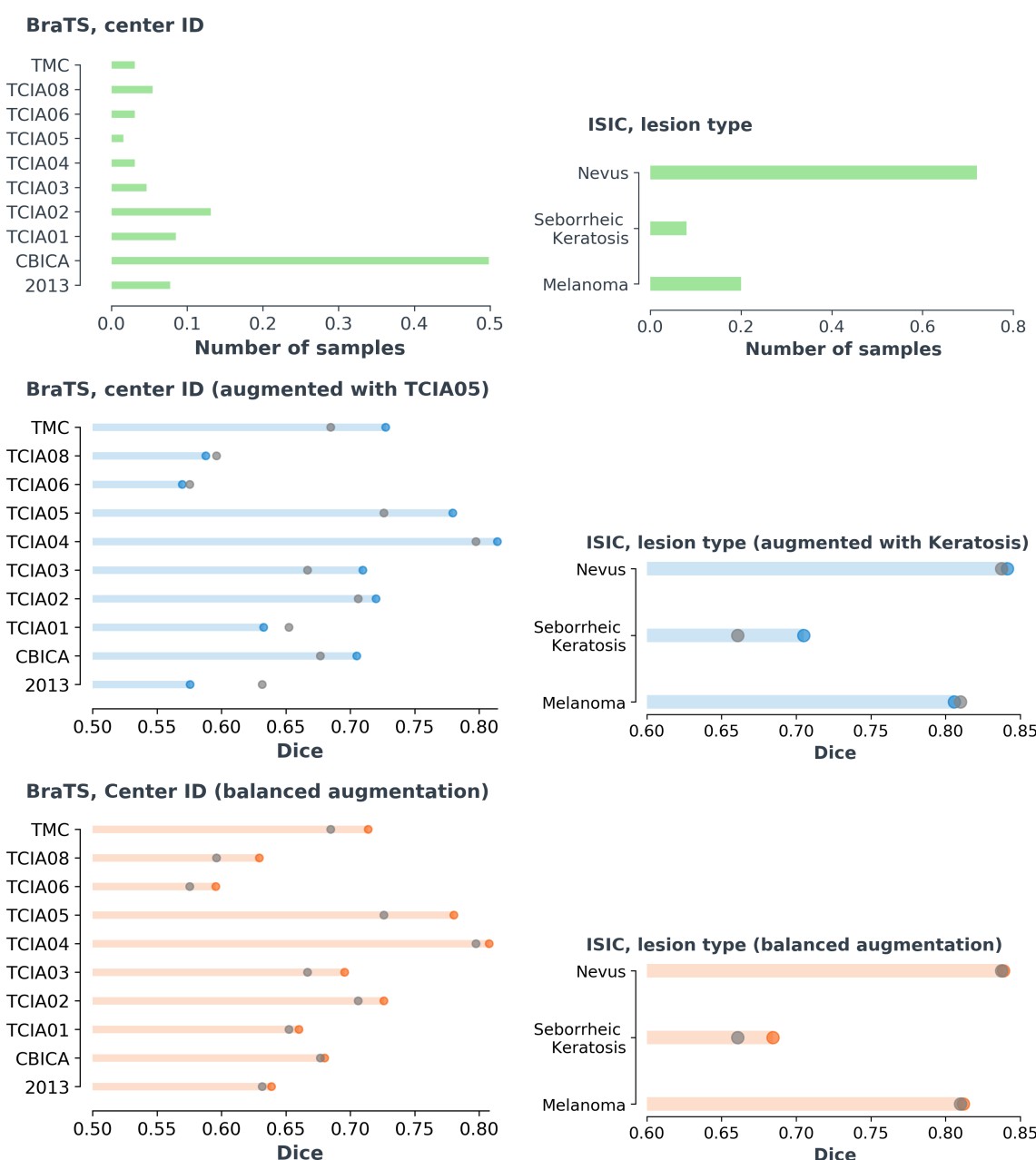

Figure 3: Class and Dice distribution over the datasets. Upper row shows distributions of the relative number of samples. In the middle row, BraTS and ISIC datasets augmented with "TCIA05" and "Seborrheic Keratosis" synthetic samples, respectively, are used for U-Net training (strategy (I)). As grey points the results for the baseline training (without the synthetic augmentation) are shown. A strong relative improvement for the injected classes suggests class-specific synthesis. Bottom row shows results for the balanced injection (strategy (II)). By means of it, we achieve increase of the DICE score for the sparse classes. The results for the mean are obtained via 3 cross-fold validation and 90/10% split for training/test data.

the same brain tumor lesion acquired from the varying acquisition environment. Thus, to learn the mask-image mapping conditioned on the lesion type is a more difficult task than to learn the mapping conditioned on the varying MRI image source. This explains poorer performance of the method on the ISIC dataset compared to BraTS.

## 4. Discussion

Datasets like BraTS and ISIC possess not only imbalance over semantic classes (tumor tissue areas/background and skin lesion area/background, respectively) but also imbalance over the global classes (i.e. center ID and lesion type). It is of practical interest to solve the imbalance problem for the latter.

If we want to use the original SPADE architecture to generate synthetic images that are global class specific in appearance in order to solve the imbalance problem, we would need to train as many different SPADE networks as there are global classes (for BraTS the number of such classes is 10) on different portions of the dataset. A single SPADE network trained on the whole dataset does not allow to control appearance. At inference time, it would generate images of either a random class or some "average" over classes appearance (potentially belonging to no global class). For example, in BraTS some centers only contain old scanners with outdated MR sequences and poor spatial resolution, while others contain recent 3T PET-MR scanners generating high resolution images. We do not want to mix textures and contrasts all across.

Using the proposed method, we only need to train a single network. Moreover, such simple conditioning allows us to make use of the representation learned during the training on the whole dataset, for the synthesis of a particular global class. To prove it quantitatively considering the TCIA05 class from BraTS (Fig.3):

- If we use the SPADE design to generate TCIA05 synthetic images, we would train the network only on the portion of the whole dataset containing TCIA05 data (which is from the whole dataset). Then, if we train a U-Net on the BraTS dataset augmented with these TCIA05 synthetic images, we achieve 0.683 accuracy.

- If we train the U-Net augmented with synthetic images generated by our proposed method we achieve 0.779 accuracy (shown in Fig.3).

In our method, we train the GAN on the whole dataset to synthesize images of a particular global class, whereas the number of real images of the global class used for Unet training is only a portion of the dataset. Thus, we say that we mitigate the "synthesis dilemma" described in the introduction.

## 5. Conclusion

We propose an architectural design for augmenting imaging data based on the generative adversarial networks. The networks conditioned on the pixel-level and global-level information allow to efficiently control visual appearance of the generated images. We show through a series of experiments that by varying class distribution of the training set (via injection of the synthetic samples into training cycle) we can regulate the accuracy levels of the datasets' global classes.

## 6. Acknowledgments

Ivan Ezhov and Suprosanna Shit are supported by the Translational Brain Imaging Training Network (TRABIT) under the EU Marie Sklodowska-Curie programme (Grant agreement ID: 765148). Anjany Sekuboyina is funded via ERC-Horizon 2020 Programme. Bjoern Menze and Florian Kofler are supported through the SFB 824, subproject B12, by Deutsche Forschungsgemeinschaft (DFG) through TUM International Graduate School of Science and Engineering (IGSSE), GSC 81, and by the Technical University of Munich – Institute for Advanced Study, funded by the German Excellence Initiative. The authors are grateful to the NVIDIA Corporation for donation of the Quadro P6000 GPU. Finally, we acknowledge Anna Valentina Lioba Eleonora Claire Javid Mamasani for eye-opening insights.

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

# Appendix A.

## A.1. Samples of BraTS synthetic images

| Masks | T1CE | FLAIR | T2 | T1 |
|---|---|---|---|---|

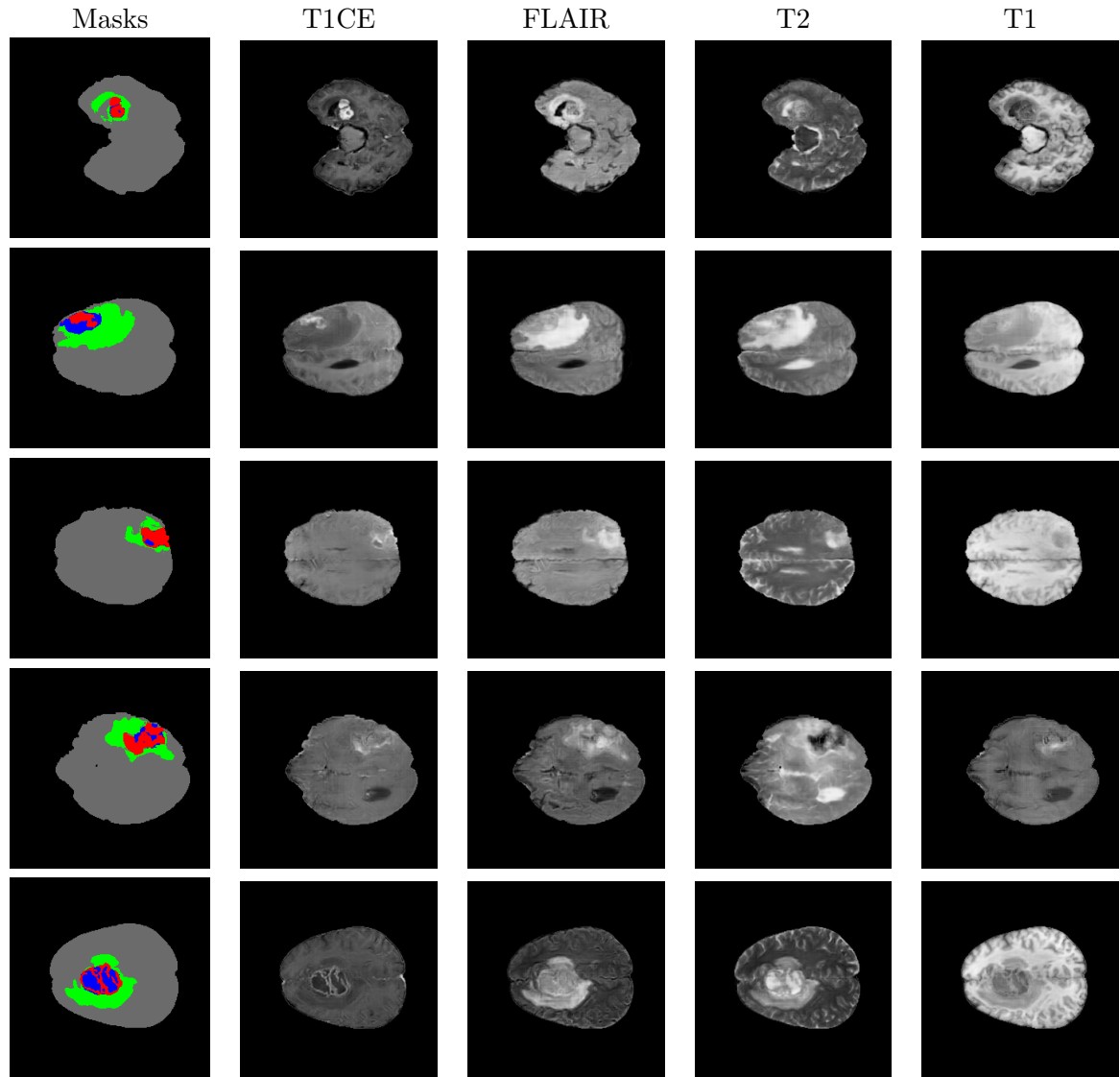

## A.2. Samples of BraTS synthetic images (class-conditioned)

| Class | T1CE | FLAIR | T2 | T1 |
|-------|------|-------|----|----|
| 2013 | | | | |
| CBICA | | | | |
| TCIA01 | | | | |
| TCIA02 | | | | |
| TCIA03 | | | | |
| TCIA04 | | | | |
| TCIA05 | | | | |
| TCIA06 | | | | |
| TCIA08 | | | | |
| TMC | | | | |

### A.3. Samples of ISIC synthetic images (class-conditioned)

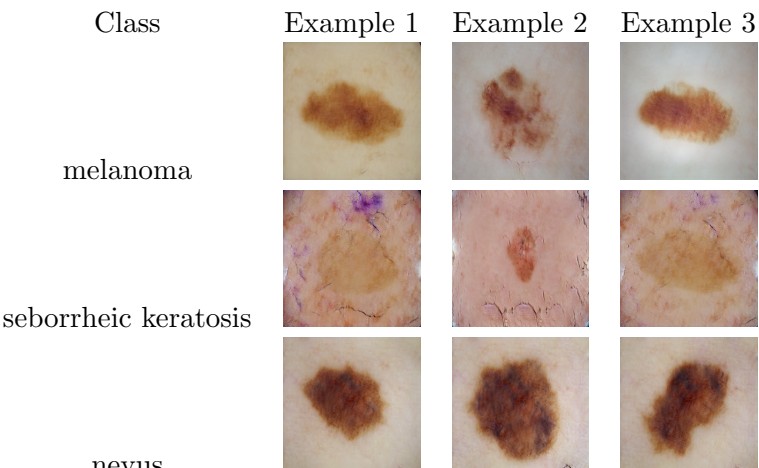

### A.4. Dice distribution for experiments with synthetic samples augmentation

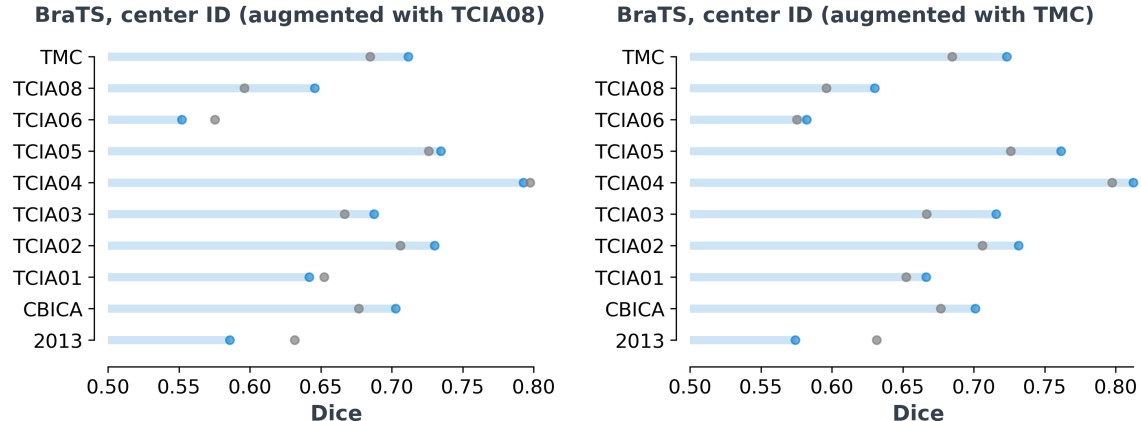

Figure 4: Dice distribution over the BraTS dataset. Synthetic samples are injected in the original set according to the strategy (I). As grey points the results for the baseline (without the synthetic augmentation) training are shown. The results for the mean are obtained via 3 cross-fold validation and 90/10% split for training/test data.

