# OpenReview forum: "Red-GAN: Attacking class imbalance via conditioned generation. Yet another medical imaging perspective"
_MIDL.io/2020/Conference — MIDL 2020_

### Official Review · AnonReviewer1 · 2020-03-11
**A data augmentation method for segmentation with GAN**

**Rating:** 3
**Confidence:** 3
**Recommendation:** Poster

**Summary:**

This paper proposed a GAN-based method for segmentation data augmentation. Specifically, the focus on the problem of class imbalance, i.e. the data imbalance of brain data from different medical centers and of skin data of different lesion types. They proposed a GAN based method to generate medical images from segmentation masks as a way of generating synthetic images. The experiments results showed improvement of segmentation with proposed data augmentation.

**Strengths:**

- Good introduction and motivation.
- Propose a GAN-based method for data augmentation.
- Generally written well.
- Adopt state-of-the-art conditioning technique (SPADE).
- Experiments on two publicly available dataset (BraTS and ISIC).

**Weaknesses:**

- Some confusions about the baseline and proposed method, e.g. what is SPADE-GAN? How it is different from proposed method?
- Method design is not clearly described and motivated.
- Lack ablation study.
- Improvements in the results are minimal without statistical significance test.
- Confusion in the description of experiment. Especially the description of single class augmentation and balance augmentation is not easy to follow.
- Lack some experiments, e.g. what are the segmentation results if you do not use synthetic images at all?

**Detailed Comments:**

- More details of the proposed method could be provided, for example, in caption of figure 2.
- Show some example result figures in the main text.


**Justification Of Rating:**

The method is interesting but it seems to lack some experiments and clarification. For example, they only compared the results with segmentation using another GAN method, but what are the segmentation results if no data augmentation is used?

**Paper Type:**

validation/application paper

**Questions To Address In The Rebuttal:**

- The numeric results do not have statistical significance test (e.g. t-test). As the some results are similar, for example the segmentation dice scores of BraTS in Table 1 are 0.6479 and 6598, it would be nice to use statistical significance test like t-test.
- It is not easy to follow what are the differences between SPADE-GAN and the proposed GAN? Could you explain more?
- What are the results if you do not use synthetic images for data augmentation?


**Special Issue:**

no

---

> ### Author Response · Authors · 2020-03-26
> **Response to reviewer 1**
>
> We thank you for reviewing the manuscript and valuable comments.
>
> “The numeric results do not have statistical significance test (e.g. t-test). As the some results are similar, for example the segmentation dice scores of BraTS in Table 1 are 0.6479 and 6598, it would be nice to use statistical significance test like t-test.”
> We agree that it is important to evaluate the statistical significance of the Dice score improvement. For the BraTS dataset in Table 1, the p-value is equal to 0.033, and for ISIC - 0.028 using the paired Wilcoxon signed rank test. Thus, the Null hypothesis can be rejected for both experiments. We will add this to the final version of the manuscript
>
> “It is not easy to follow what are the differences between SPADE-GAN and the proposed GAN? Could you explain more?”
> We made two major modifications to the original SPADE architecture. First, is the global conditioning, and second is the inclusion of the segmentor. We thank you for pointing out that this may not be entirely clear and will make sure that this gets conveyed more clearly in the final version
>
> “What are the results if you do not use synthetic images for data augmentation?”
> In Figure 3, the accuracy for the baseline without injection of the synthetic images are depicted as grey dots. We will make it more clear in the final version.
>
> "Show some example result figures in the main text."
> We agree with you, a plot with qualitative comparison in the main text would be of use for a reader.

---

### Official Review · AnonReviewer4 · 2020-03-12
**Very interesting work with minor issues**

**Rating:** 3
**Confidence:** 5

**Summary:**

In this work, the authors modified the SPADE framework by adding class-wise information (protocol, vendor, etc.) to synthesize medical images for different modalities/protocols. They add a pre-trained U-net segmentor to constrain the synthesized images in the proximity of real images. The result shows improvement in the segmentation task by adding synthesized images. This work would probably mitigate data scarcity in the medical field by cross-class image synthesis.

**Strengths:**

1. Very natural migration from a single condition (SPADE) to a dual conditional GAN
2. The U-net segmentor added in the framework helps to maintain the constancy is an interesting idea
3. Clear result presentation and sufficient validation.


**Weaknesses:**

The method part is not very clearly written.
1. The authors do not show how the segmentor plays in the role of constraint the synthesis.
2. The GAN loss part is not very clearly explained.
3. No explanation on the transformation of a mask to a 1024x8x8 tensor


**Detailed Comments:**

1. In the context, ‘segmentor’ is used but in the graph ‘segmentator’ is used. Please stick to one word.
2. The generator input (random noise) might as well be plotted in the graph.
3. It would be nice if the authors can include some validation from medical experts to show the realness of generated images.


**Justification Of Rating:**

The major contribution of this work is the ‘global’ information utilized and the U-net segmentation network used in the GAN.  It contributes to cross-modality/protocols image synthesis which mitigates the data scarcity greatly. The constraint adds to the image synthesis could be a double-edged sword since it balances between the similarity and variability of image synthesis. This work would be very promising if it shows more variability in image synthesis.

**Paper Type:**

validation/application paper

**Questions To Address In The Rebuttal:**

1. Not clear in the GAN loss part, which by my personal assumption it would be the same as vanilla GAN. If not, please clarify.
2. Not clear how the feature output of the segmentor plays in the role of constrained synthesis. Does it also involved in the objective? Please explain.
3. No explanation in figure 1 and 2. Please add some for a better understanding.
4. Please explain the non-linear transform is applied from mask to a 1024x8x8 tensor.
5. If possible, it would also be nice to show the variability of image generation. How not unlike the image synthesized would be different from the real image. Or if this kind of operation is limited since this framework constrained the synthesis to maintain consistency.


**Special Issue:**

no

---

> ### Author Response · Authors · 2020-03-26
> **Response to reviewer 4**
>
> We thank you for reviewing the manuscript and valuable comments.
>
> “Not clear in the GAN loss part, which by my personal assumption it would be the same as vanilla GAN. If not, please clarify.”
> We used Hinge loss and feature matching loss analogous to the SPADE design (we base our work on).
> SPADE paper -  https://arxiv.org/pdf/1903.07291.pdf
> Related Pix2PixHD paper on which the SPADE is based - https://arxiv.org/pdf/1711.11585.pdf
>
> “Not clear how the feature output of the segmentor plays in the role of constrained synthesis. Does it also involved in the objective? Please explain.”
> The segmentor's addition does not affect the class-specific synthesis and is motivated by our desire to use the synthetic images for segmentation task. If we train a generator from scratch, then it learns a general representation of images. Since we want to use the synthetic images for segmentation, we want to ensure the images lie within close proximity to the real images in the latent representation, based on which the segmentor makes its decision.
> The segmentor does not affect the objective function. The feature maps from the segmentor are only concatenated with the discriminator's input.
> It is the global class conditioning (details of which we explain below) that controls synthetic image appearance.
>
> “No explanation in figure 1 and 2. Please add some for a better understanding.
> Please explain the non-linear transform is applied from mask to a 1024x8x8 tensor.”
> Thank you for pointing it out. For the transformation of the mask we used downsampling to 8x8 size, followed by a convolution operation with 1024 filters. Whereas for the global label conditioning, we used an embedding layer followed by a fully connected layer scaling the embedding to the 1024x8x8 size before concatenating it in the generator. This way of conditioning is analogous to:
> - a highly cited NeurIPS paper:
> https://papers.nips.cc/paper/5773-deep-generative-image-models-using-a-laplacian-pyramid-of-adversarial-networks.pdf
> - a list of “hacks” to train Gans from facebook/deepmind researchers (p. 16):
> https://github.com/soumith/ganhacks
> We will make it more clear in the final version of the manuscript and release the code.

---

> > ### Comment · AnonReviewer4 · 2020-04-02
> > **Thanks for the response.**
> >
> > The explanation helps much.  I think it is a well-executed paper. However, I cannot change my rating in the reviewing panel. If the chairs are checking, I hope I can re-state my opinion on this paper. I would like to change my rating from 3 to 4 strong accept as a poster.

---

### Official Review · AnonReviewer3 · 2020-03-13
**Another study that shows GAN generated images can help mitigate the class imbalance problem**

**Rating:** 2
**Confidence:** 4
**Recommendation:** Poster

**Summary:**

The paper proposed to use SPADE to perform conditioned image generation to cope with class imbalance problem. The information the generator conditions includes both the local one which is the segmentation mask and the global one which can be the acquisition center or lesion type. A segmentor is further incorporated to the architecture which claimed to help the downstream segmentation task.

**Strengths:**

1. The paper is well written and easy to follow
2. I'm glad the authors have provided a lot of training details
3. The review of the related works section is satisfactory
4. The centre ID conditioned generation will be of interest to industry

**Weaknesses:**

The main weakness I think is the experiment results are not enough to justify the claimed contributions. I do not see how the proposed local-global conditioning allows to mitigate the "synthesis dilemma" as claimed in the paper. The global information, e.g. acquisition center is essentially a special case of mask where all pixels inside the mask share the same value. Does the original SPADE method not able to handle this?


**Detailed Comments:**

1.In Figure 3, the caption of the first two graphs should be percentage of samples.
2. What is the GAN loss, Hinge, least square or others?
3. What is the receptive field for each sub-discriminator?
4. What is the non-linear mapping for the injection of the class information?
5. Where does  Rubeus in the name referring to?
6. Why the discriminator is not conditioned on the global label like the setup of ACGAN/CGAN?

**Justification Of Rating:**

There are many works out there that claim the GAN generated images are beneficial for the training.  The proposed work just used a more recent approach to perform the generation. I don't see any clear evidence that the segmentor is actually contributing to the improvement. I'm giving a weak rejection for now. If the authors can provide concreate evidecne, I'm more than happy to change my rating.

**Paper Type:**

validation/application paper

**Questions To Address In The Rebuttal:**

1. What is the setup of the original SPADE-GAN and what are the exact differences between the SPADE-GAN and the proposed Rubeus-GAN? It looks to me that the authors have made two adjustments based on the original SPADE architecture. First is the global conditioning and second is the segmentor.   I would expect to see an ablation study on the usefulness of each of the addons.
2. What are the inputs for the training of the segmentor? Has the segmentor get well trained with 100 epochs for both experiments. I do not see any validation on the performance of the segmentor.
3. Are there any other type of augmentation been used in the training of the downstream segmentation network? If yes, then what are those; if not, please provide the justification.


**Special Issue:**

no

---

> ### Author Response · Authors · 2020-03-26
> **Response to reviewer 3**
>
> Thank you for giving us very constructive feedback.
> “I do not see how the proposed local-global conditioning allows to mitigate the "synthesis dilemma" as claimed in the paper”
> As you correctly pointed out in the summary, one of the contributions of our methodology is introducing a segmentor into the two-player game. However, the global label conditioning that allows to control appearance of the generated images is even more important. We will try to explain it here in greater details.
> Datasets like BraTS and ISIC possess not only imbalance over data classes (tumor tissue areas/background and skin lesion area/background, respectively) but also imbalance over the global classes (i.e. center ID and lesion type). It is of practical interest to solve the imbalance problem for the latter.
> If we want to use the original SPADE architecture to generate synthetic images that are global class specific in appearance in order to solve the imbalance problem, we would need to train as many different SPADE networks as there are global classes (for BraTS the number of such classes is 10) on different subportions of the dataset. A single SPADE network trained on the whole dataset does not allow to control appearance. At inference time, it would generate images of either a random class or some "average" over classes appearance (potentially belonging to no global class). For example, in BraTS some centers only contain rather old scanners with outdated MR sequences and poor spatial resolution, while others contain recent 3T PET-MR scanners generating high resolution images. We don't want to mix textures and contrasts all across.
> Using the proposed method, we only need to train a single network. Moreover, such simple conditioning allows us to make use of the representation learned during the training on the whole dataset, for the synthesis of a particular global class. To prove it quantitatively considering the TCIA05 class from BraTS (from Fig.3):
> - if we use the SPADE design to generate TCIA05 synthetic images, we would train the network only on the portion of the whole dataset containing TCIA05 data (which is $<5\%$ from the whole dataset). Then, if we train a U-Net on the BraTS dataset augmented with these TCIA05 synthetic images, we achieve 0.683 accuracy (this number was not shown in the manuscript and was computed to support our argument).
> - If we train the U-Net augmented with synthetic images generated by our proposed method we achieve 0.779 accuracy (shown in Fig.3).
> In our method, we train the GAN on the whole dataset to synthesize images of a particular global class, whereas the number of real images of the global class used for Unet training is only a portion ($<5\%$ TCIA05 case) of the dataset. Thus, we say that we mitigate the "synthesis dilemma" described in the introduction.
> We hope that the benefit is now more clear and will make sure that the final version of the manuscript conveys this message.
>
> “What are the inputs for the training of the segmentor? Has the segmentor get well trained with 100 epochs for both experiments. I do not see any validation on the performance of the segmentor”
> The segmentor is trained on the original datasets till 100 epochs. Symptoms of convergence are present for both experiments. In fig. 3 the grey dots depict the baseline segmentor performance.
>
> “Are there any other type of augmentation been used in the training of the downstream segmentation network? If yes, then what are those; if not, please provide the justification”
> We didn't apply any other augmentations because our aim was to test the viability of our contributions i.e. the segmentor and global conditioning on the downstream segmentation task without introducing other factors which can affect the segmentation results
>
> “In Figure 3, the caption of the first two graphs should be percentage of samples
> What is the GAN loss, Hinge,least square or others?
> What is the receptive field for each sub-discriminator?"
> Thank you for pointing it out. We will make it more clear in the final version and release the code.
>
> “Where does Rubeus in the name referring to?”
> The original intuitive title of the paper was "Red-Gan: attacking class imbalance..." that we changed realizing it can provoke some non-scientific concerns. The word "Rubeus" refers to "red" in Latin. We may change the title back to avoid any confusion ;)
>
> "Why the discriminator is not conditioned on the global label like the setup of ACGAN/CGAN?"
> We agree that there are other ways to incorporate the class label. In our case, we used an embedding layer followed by a fully connected layer scaling the embedding to the 1024x8x8 size before concatenating it in the generator. This way of conditioning is analogous to:
> - a highly-cited NeurIPS paper:
> https://papers.nips.cc/paper/5773-deep-generative-image-models-using-a-laplacian-pyramid-of-adversarial-networks.pdf
> - a list of “hacks” to train GANs from facebook/deepmind researchers (p. 16):
> https://github.com/soumith/ganhacks

---

### Official Review · AnonReviewer2 · 2020-03-15
**Reasonable Methodology but Insufficient Experiments**

**Rating:** 3
**Confidence:** 5
**Recommendation:** Poster

**Summary:**

This paper extends the SPADE-GAN framework for generating images from masks. In particular, it involves a segmentation network in the conventional two-player game. The segmentation network works as a "representation" enhancement for the generated images. Combing the enhanced "representation" (which is actually a segmentation map) and the real/generated masks, the discriminator can learn better.

**Strengths:**

1. The methodology introducing a segmentor to the two-player game is reasonable. I agree with the argument "If we train a generator from scratch, then it learns a general representation of images. Since we want to use the synthetic images for segmentation task, we want to ensure the images lie within close proximity to the real images in the latent representation, based on which the segmentor makes its decision."
2. Very detailed description of the method.
3. The method seems promising though improvement of Dice is limited compared to SPADE-GAN.

**Weaknesses:**

1. The experiments are insufficient. At lease, I'd like to see how is the quality of the generated images through direct evaluation (segmentation is an indirect evaluation).
2. No analysis. What's the cause of the performance gain towards SPADE-GAN? Because of the introduced segmentor or the new network arch or training ....?
3. No standard deviation provided.
4. How is your global label contribute to the method? Analysis in experimental section?

**Justification Of Rating:**

1. The methodology is reasonable and I agree with their argument.. But the story is a little bit inconsistent. Since the authors emphasize the global label, while I am not convinced that the global information in the generator works or solve the problem they mentioned.
2. The provided experiments are really not sufficient.

**Paper Type:**

methodological development

**Questions To Address In The Rebuttal:**

Please answer the questions I mentioned in the weakness.

**Special Issue:**

no

---

> ### Author Response · Authors · 2020-03-26
> **Response to reviewer 2**
>
> We thank you for reviewing the manuscript and valuable comments.
>
> “How is your global label contribute to the method? Analysis in experimental section?”
> As you correctly pointed out in the summary, one of the contributions of our methodology is introducing a segmentor into the two-player game. However, the global label conditioning that allows to control appearance of the generated images is even more important. We will try to explain it here in greater details.
> Datasets like BraTS and ISIC possess not only imbalance over data classes (tumor tissue areas/background and skin lesion area/background, respectively) but also imbalance over the global classes (i.e. center ID and lesion type). It is of practical interest to solve the imbalance problem for the latter.
> If we want to use the original SPADE architecture to generate synthetic images that are global class specific in appearance in order to solve the imbalance problem, we would need to train as many different SPADE networks as there are global classes (for BraTS the number of such classes is 10) on different subportions of the dataset. A single SPADE network trained on the whole dataset does not allow to control appearance. At inference time, it would generate images of either a random class or some "average" over classes appearance (potentially belonging to no global class). For example, in BraTS some centers only contain rather old scanners with outdated MR sequences and poor spatial resolution, while others contain recent 3T PET-MR scanners generating high resolution images. We don't want to mix textures and contrasts all across.
> Using the proposed method, we only need to train a single network. Moreover, such simple conditioning allows us to make use of the representation learned during the training on the whole dataset, for the synthesis of a particular global class. To prove it quantitatively considering the TCIA05 class from BraTS (from Fig.3):
> - if we use the SPADE design to generate TCIA05 synthetic images, we would train the network only on the portion of the whole dataset containing TCIA05 data (which is $<5 \%$ from the whole dataset). Then, if we train a U-Net on the BraTS dataset augmented with these TCIA05 synthetic images, we achieve 0.683 accuracy (this number was not shown in the manuscript and was computed to support our argument).
> - If we train the U-Net augmented with synthetic images generated by our proposed method we achieve 0.779 accuracy (shown in Fig.3).
> In our method, we train the GAN on the whole dataset to synthesize images of a particular global class, whereas the number of real images of the global class used for Unet training is only a portion ($<5 \%$ TCIA05 case) of the dataset. Thus, we say that we mitigate the "synthesis dilemma" described in the introduction.
> We hope that the benefit is now more clear and we will make sure that the final version of the manuscript conveys this message even more clearly.
>
> “The experiments are insufficient. At lease, I'd like to see how is the quality of the generated images through direct evaluation (segmentation is an indirect evaluation).”
> The goal of our work was to bypass the “synthesis dilemma”: to exploit the GAN-based augmentation of the original training set for the downstream segmentation task. Thus, the evaluation criteria, we were mainly interested in, was the segmentation accuracy. We placed the images due to paper size restrictions into the appendix, but we agree a plot with qualitative comparison in the main text would be of use for a reader.
>
> “What's the cause of the performance gain towards SPADE-GAN? Because of the introduced segmentor or the new network arch or training ....?”
> What concerns table 1, the performance gain towards SPADE-GAN is only due to the introduction of the third player into the adversarial game. All the training and architectural details were kept the same.
>
> “No standard deviation provided.”
> We agree with you and will add the data to the final version of the manuscript.

---

### Meta-Review · Area_Chair1 · 2020-04-05
**MetaReview of Paper71 by AreaChair1**

**Rating:** 3
**Recommendation For Accepted Papers:** Poster

**Metareview:**

Overall, the paper appears to be well written and present an interesting development based on previous work but the lack in quantitative validation is consistently highlighted

**Paper Type:**

methodological development

**Special Issue:**

no

---

> ### Author Response · Authors · 2020-04-09
> **Response to Area Chair**
>
> Dear Area Chair,
>
> thank you for the positive review on our paper.
>
> Regarding the quantitative validation, we addressed all concerns of the reviewers in the rebuttal. We have two main contributions: introducing a third player (segmentor) into the two-player adversarial game and global class conditioning. Ablation studies on the usefulness (accuracy gain) of each of the contributions are provided in the method part. We will present it more clearly in the final paper version.

---

### Decision · Program_Chairs · 2020-04-11

Accept